# Elemental Selenium Enriched Nanofiber Production

**DOI:** 10.3390/molecules26216457

**Published:** 2021-10-26

**Authors:** Khandsuren Badgar, József Prokisch

**Affiliations:** 1Institute of Animal Science, Biotechnology and Nature Conservation, Faculty of Agricultural and Food Sciences and Environmental Management, University of Debrecen, 138 Böszörményi Street, 4032 Debrecen, Hungary; jprokisch@agr.unideb.hu; 2Doctoral School of Animal Science, University of Debrecen, 138 Böszörményi Street, 4032 Debrecen, Hungary

**Keywords:** selenium nanoparticles, nanopowder, nanofibers, electrospinning

## Abstract

This study aimed to produce electrospun nanofibers from a polyvinyl butyral polymer (PVB) solution enriched with red and grey selenium nanoparticles. Scanning electron microscopic analysis was used to observe the samples, evaluate the fiber diameters, and reveal eventual artifacts in the nanofibrous structure. Average fiber diameter is determined by manually measuring the diameters of randomly selected fibers on scanning electron microscopic (SEM) images. The obtained nanofibers are amorphous with a diameter of approximately 500 nm, a specific surface area of approx. 8 m^2^ g^−1^, and 5093 km cm^−3^ length. If the red and grey selenium nanoparticles were produced in powder form and suspended to the ethanolic solution of PVB then they were located inside and outside the fiber. When selenium nanoparticles were synthesized in the PVB solution, then they were located only inside the fiber. These nanofiber sheets enriched with selenium nanoparticles could be a good candidate for high-efficiency filter materials and medical applications.

## 1. Introduction

Electrospun nanofibers have attracted great attention owing to their unique features such as high specific surface areas, high porosity, and interconnected pore structure. Due to these properties, they have been used infiltration [1,2], nanocomposites [3,4], biological functional tissue scaffolds [5,6,7], protective textiles [8,9], catalysis [10,11], drug delivery systems [6,12], etc. The electrospinning process is a straightforward experimental setup that spins diameters ranging from 10 nm to several hundred nanometers. The nanofibers have been successfully produced by the electrospinning technique from many types of synthetic and natural polymers. Electrospinning allows the fabrication of ultrathin fiber mats with an extraordinary control of their structure and properties. It is an ideal alternative for applications such as wound healing or even functional membranes. Control of reactive oxygen species (ROS) at the injury site plays a major role in the wound healing process. This multi-step process includes hemostasis, inflammation, proliferation, and maturation phases [13,14]. A high level of ROS could damage normal cells during the inflammatory phase, which is released from neutrophils cells to fight microbes [15,16]. In this case, applying antioxidant agents could accelerate wound healing by reducing the adverse effects of ROS [17]. Based on these, functional nanofibers have been engaging in the production of electrospun nanofibrous membranes. Recently, the production of functional electrospun nanofiber with additives, especially nanoparticles with therapeutic and preventive effects, has been a new challenge. There are few studies reported that silver nanoparticles (AgNPs) [18,19], copper nanoparticles (CuNPs) [20], gold nanoparticles (AuNPs) [21], cerium oxide nanoparticles (CeNPs) [22], selenium nanoparticles (SeNPs) [3,23] have been used for the fabrication of nanomaterials based on their antioxidant and antibacterial properties. Definitely, selenium nanoparticle is a good candidate for the development of functional properties of nanomaterials according to its high antioxidant [24], anticancer [25,26,27], detoxification [28,29], antibacterial [30,31,32,33], antifungal [34,35], and antiviral effects [36]. However, a limited number of studies were found involving electrospun nanofibers functionalized by selenium nanoparticles [3,23,37,38]. Chung and co-workers reported that silk scaffolds functionalized by selenium nanoparticles were significantly increased bacterial inhibition. In other words, it was dramatically improved the short-term human dermal fibroblast metabolic activity while reducing the ATP content of *Staphylococcus aureus* [3]. A recent study reported the effect of nanofiber functionalized by selenium nanoparticles in the application of wound healing. Nanofibers with selenium nanoparticles and vitamin E produced from polycaprolactone/gelatin have supported the proliferation and attachment of fibroblast 3T3 cells by reducing edema, inflammation, and oxidative stress at the site of injury [23]. This study tested red and grey selenium nanoparticles with polyvinyl butyral (PVB) polymer to produce nanofibers by the electrospinning method. PVB is one of the polymers that have been extensively used in many applications since PVB is a low-cost alternative, showing flexibility, optical clarity, and good adhesion to many surfaces. Furthermore, the PVB is insoluble water; therefore, it is possible to filtrate aqueous solutions. However, few studies have been performed to date on the usage of PVB in the fabric of electrospun nanofibers [39,40,41,42,43]. In addition, few cell cultures studies have been performed on this material and were based on films made by solvent casting and nanoparticles as prospective candidates in alveolar bone substitutes, drug delivery systems, or contrasting agents in cancer therapy [44,45].

## 2. Results and Discussion

In electrospinning, the process parameters (type of polymer and solvent, viscosity, concentration, net charge density, and surface tension of the polymer fluid) and the system parameters (voltage, flow rate of polymer solution, distance between capillary end and collector, ambient parameters, and motion of collector) have a crucial role in fiber morphology [46]. In our study, ten mL h^−1^ of flow rate, 25 cm of distance between stainless-steel collector and needle tip, 40 kV of voltage, 10% of PVB polymer dissolved in ethanol were compatible in the electrospinning process at the humidity 34% and temperature 24 °C. Also, the viscosity and density of PVB were found 474 mm^2^ s^−1^, 0.81 g mL^−1^, respectively. It is reported that all solutions based on isopropanol, butanol, and ethanol were suitable solvents for PVB in the production of nanofiber [39]. Ten percent of PVB polymer was suitable for resuspension selenium nanopowder, as presented in Figure 5. First, the morphology of nanofiber was checked under a simple light microscope with 400× magnification. A few layers of nanofiber were covered on the coverslip. Nanofibers have cylinder shapes with multi structures such as linear, spiral, zigzag lines, etc. as shown in Figure 1. It was a simple way of checking the morphology and structure of nanofibers.

Up to 10% of red and grey selenium nanopowder contained PVB polymer solution was suitable for nanofiber formation. The color of the fiber sheet has also increased with the increment of the concentration of selenium nanopowder. 10% of red and grey selenium nanopowder resulted in brighter pink and black nanofiber (Figure 2). There was no difference in the nanofibers’ diameter size or morphology and electrospinning process between amorphous and crystalline forms of selenium. The surface of all-fiber sheets was contained various concentrations of selenium nanopowder that were smooth and flexible. However, their high concentrations were affected by the viscosity, density, and conductivity of polymer solution. Thus, the production rate slowed down.

The morphology and diameter of red and grey selenium nanopowder enriched nanofiber was determined by scanning electron microscope with X-ray and EDS (Figure 2). The selenium nanoparticle’s beads were loaded inside and outside the nanofibers confirmed by X-Ray and EDS analysis. Generally, it is an essential technique for the observation of nanofibrous structures. As a result, increasing the concentration of nanopowder from 1 to 10% (*w*/*v*) resulted in the increased formation of beads. Similar to our finding, Kamaruzaman et al. reported that the formation of beads increases by the high concentration of selenium nanoparticles in the polymer [38]. Particles are structured at different scales due to solid interaction, resulting in chain-like structures. The aggregation of nanoparticles develops the bead formation during the electrospinning process, which causes a higher relative surface area and a higher relative number of surface atoms. Thus, the advantages of bead formation are to be load more selenium in sheet and increase its releasing and positive effects on the surface. Due to these advantages, red and grey selenium nanopowder embedded nanofibers are more suitable for filtration design in water and air, and the wound healing process. However, the chain-like structure of beads may affect fiber’s strength. The obtained nanofibers are prevalently amorphous with diameters ranging from 100 to 1000 nm and a specific surface area of approximately 4–40 m^2^ g^−1^. The average diameter of fibers was about 500 nm, and specific surfaces area 8 m^2^ g^−1^ for all selenium nanopowder concentrations. The fiber’s length was calculated from their diameter based on the equation for the volume and surface of cylinders. In this case, 1 mL of 10% polymer solution was originated from 500 nm wide nanofiber, their length was 5093 km cm^−3^.

In the next section, selenium nanoparticles were synthesized in the PVB polymer after then nanofibers were produced. In a typical procedure, 500 mg dm^–3^ of sodium selenite and 10 g dm^–3^ of ascorbic acid were dissolved in 80% of ethanol by ultrasonication for 30 min. Then, 10% of PVB was added to each, and the mixture was ultrasonicated for 30 min. The sodium selenite and ascorbic acid-enriched PVB were kept at room temperature for a day, separately too. On the next day, these solutions were mixed in the same ratio at ultrasonic for 30 min (SeNPs-PVB). At the end of the reaction time, the color of the solution was turned red. In the final step, SeNPs-PVB solution was diluted with pure PVB solution in the same ratio (shown in Figure 5C). Because 80% of ethanol was not good for electrospinning, it was changed conductivity. The reduction process was similar to the previous experiment regarding reaction time and color change [47]. Our earlier study found that selenium nanoparticles aggregate to a large extent in a liquid, and the size of the aggregates grew up to 100 µm by reaction time [48]. Smaller particles readily interact with each other to form larger particle sizes [49]. The agglomeration occurs when surface atoms of nanoparticles tend to form bonds due to the smaller size of the nanoparticles. Nevertheless, PVB polymer solution does not include any large aggregate of selenium nanoparticles, and it was a good stabilizer. Also, PVB is completely non-toxic. Due to its composition of only carbon, hydrogen, and oxygen, it combusts with almost no residue, which resulted in its widespread use in food packaging. As a result, we have a good fiber sheet with light pink, smooth and soft, which is contained 1250 mg dm^−3^ of selenium (Figure 3). A scanning electron microscope also characterized the morphology, diameter, and location of selenium nanoparticles. The selenium nanoparticles, which are well dispersed in the PVB solution, played an important factor in the fabrication of high-quality nanofibers. There are no particles on the outside of the fiber that was indicated by electron microscopic images (Figure 3). The absence of selenium nanoparticles on the outside of the fiber was visible under an electron microscope. All the selenium nanoparticles with a diameter of 50 nm or smaller are located on the inside of the fiber. The mean diameter of fibers was 500 nm. Their specific surface area and length are also similar to the previous results. It was our hypothesis for this section of the experiment. This result was confirmed that any selenium nanoparticles do not affect the morphology of the nanofibers. The produced fiber sheets are new material. Also, it is a new challenge for the production of functional nanofiber. The functional nanofiber, in particular, has attained a greater interest in recent years. The applications of functional nanofibers are increasing in various technical fields including filtration [50,51], tissue engineering [3,7], drug delivery systems [12], wound dressings [21,23], and antibacterial [18,22]. Thus, selenium nanoparticles enriched nanofibers could be good material for medical technology in improving wound healing and reducing infection without antibiotics. Especially, it has the potential to be applied in skin tissue and filtration. From the previous study, selenium nanoparticles have shown antioxidant [24], anticancer [25,26,27], antibacterial [30,31,32,33], and antifungal properties [34,35]. For instance, the addition of selenium nanoparticles in nanomaterials improved fibroblast’s metabolic activity while reducing the ATP content of *S. aureus* [3] and completed re-epithelization [23] for skin application.

## 3. Materials and Methods

Sodium selenite (Na_2_SeO_3_) and ascorbic acid (C_6_H_8_O_6_) were purchased from VWR International Ltd. (Lutterworth, Leics. UK). Ethanol (ethyl alcohol) was purchased from Nógrádi Vegyipari Zrt (Tolmács, Hungary). Polyvinyl butyral (PVB, Mowital LPB 16H, molar mass 16 kDa) was purchased from Kuraray Europe GmbH D-65926 (Frankfurt am Main, Germany).

### 3.1. Preparation of Selenium Nanoparticles

A modified green chemical reduction method was used for the synthesis of selenium nanoparticles [47].

Selenium nanogranules were prepared according to our previous method [48]. Briefly, 10,000 mg dm^–3^ of selenium in the form of sodium selenite solution and 100 g dm^–3^ of ascorbic acid was mixed at the same ratio, and the mixture was kept at room temperature for 2 h. At the end of the reaction, nanogranules were filtered by a filter paper and washed with alcohol and distilled water 3 times. The red nanogranules were dried at 4 °C for several days (Figure 4). Grey hexagonal selenium nano granules were prepared from the dried red selenium suspension keeping it at the temperature of 85 °C for 10 min. Finally, red and grey selenium nano-granules were powdered by a nano grinder. These allotropes of selenium were identified by X-ray diffraction analysis (Figure 4).

### 3.2. Preparation of SeNPs-PVB Solution

Ten percent of polyvinyl butyral (PVB) solution was dissolved in absolute ethanol under constant stirring at room temperature until the solution was transparent. The solutions were left standing without stirring for at least a day. Three types of selenium-enriched polymer solution were prepared for the experiment. Two solutions contained the previously prepared red and grey selenium nanopowder. Various red and grey nanopowder (1, 3, 5, 10% *w*/*v*) were infused in the PVB solution and sonicated for 30 min (Figure 5A,B). The third polymer solution was prepared a different way. The sodium selenite and the ascorbic acid were dissolved in 80% of ethanol-20% of water solution separately, then 10% of PVB polymer was dissolved After dissolution, the two solutions were mixed in a 1:1 ratio at room temperature to produce selenium nanoparticles (Figure 5C).

### 3.3. Electrospinning

PVB polymer solution was enriched with 2 types of selenium nanoparticles at various concentrations and was used to produce nanofibers by horizontal electrospinning setup. The polymer solution was loaded in a 50 mL syringe and injected through a stainless-steel blunt-ended needle at a flow rate of 10 mL h^−1^. The nanofibers were collected on the stainless-steel collector placed at a distance of 25 cm from the tip of the needle under a voltage of 40 kV (voltage power supply NT-45/P, RWT Vasúttechnikai Kft) at the humidity (RH) 34% and temperature at 24 °C. The length of fiber is calculated according to the following Equation (1):(1)$$l=\frac{4V}{d^{2}\pi }$$
where “*l*” is the length of fiber in meter, “*V*” is the volume in m^3^, “*d*” is the diameter in meter.

### 3.4. Specimen Characterization

Morphological and microstructural features of the fibers were investigated by field-emission scanning electron microscopy (SEM), by using a Hitachi S-4300 with Energy-dispersive X-ray (EDS) analysis, and simple light microscope (Visiscope^®^ 260, VWR). The samples were collected on glass slides and textile covered stainless-steel collector for characterization.

## 4. Conclusions

Our work studied the presence of selenium nanoparticles in powder and aqueous for the enrichment of electrospun nanofiber. The average size of originated fibers is approx. 500 nm, specific surface area 8 m^2^ g^−1^, and their length was 5093 km cm^−3^ for all used selenium nanoparticles were obtained. In our study, we obtained that the allotropes of selenium have no effects on the morphology and diameter of the nanofibers. The different selenium allotropes would have a difference in the solubility and the adsorption potential of mercury vapor. The presence of selenium nanopowder distributed beads on the outside and inside of the fiber. Increasing the concentration of selenium nanopowder has increased the formation of beads. When selenium nanoparticles are synthesized in the PVB solution, then the selenium nanoparticles are inside the fiber, the size is much smaller than 50 nm. Thus, the originated fibers were formed with uniform diameter and smooth surface. Further works are required to study the efficiency of filters produced from the selenium-enriched PVB nanofibers.

## Figures and Tables

**Figure 1 molecules-26-06457-f001:**
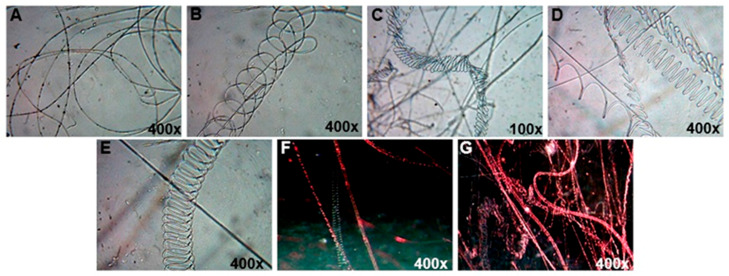
The multiple structures of nanofibers enriched by selenium nanoparticles; (**A**,**B**): red selenium nanopowder, (**C**): grey selenium nanopowder, (**D**–**G**): aqueous-ethanol SeNPs. The light microscopy pictures were made with white light and red laser illumination.

**Figure 2 molecules-26-06457-f002:**
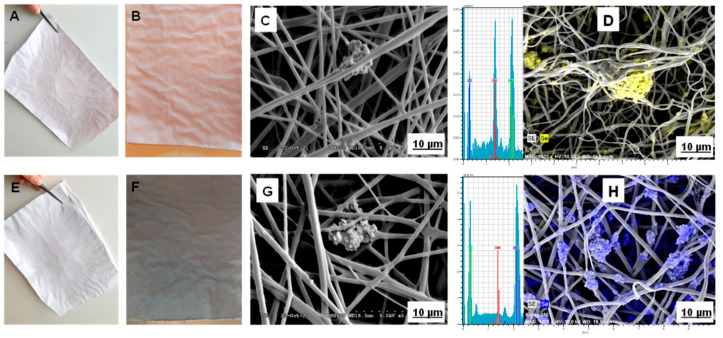
Nanofiber sheets enriched by selenium nanopowder and their SEM images with EDS; (**A**,**C**): 1% red selenium, (**B**,**D**): 10% red selenium, (**E**,**G**): 1% grey selenium, (**F**,**H**): 10% grey selenium.

**Figure 3 molecules-26-06457-f003:**
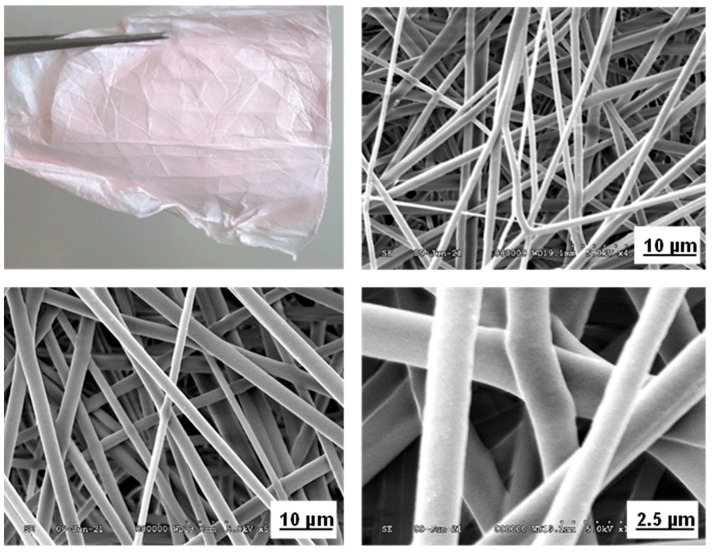
Nanofiber sheet enriched by aqueous-ethanol SeNPs and its SEM images.

**Figure 4 molecules-26-06457-f004:**
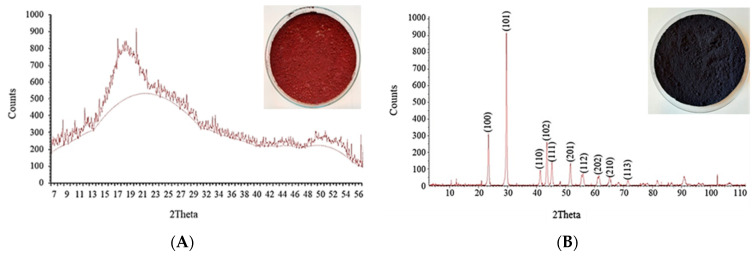
XRD pattern with red (**A**) and grey (**B**) selenium nanopowder.

**Figure 5 molecules-26-06457-f005:**
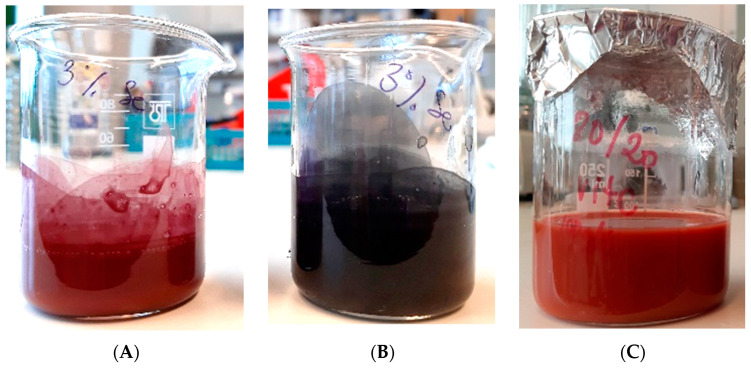
PVB polymer solution enriched with red (**A**), grey selenium nanopowder (**B**), and aqueous-ethanol SeNPs (**C**).

## Data Availability

The data presented in this study are available on request from the corresponding author.

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
