# Peer review of "Elemental Selenium Enriched Nanofiber Production"

_molecules, 2021, doi:10.3390/molecules26216457_

Round 1
Reviewer 1 Report
This article introduces a method of using Electrospinning to enrich SeNPs and synthesis relevant nanofiber. Generally speaking, the article has some innovation and value, but from the perspective of the full text, there should be further improvements focusing on the research, especially on the application potentials. So I do not recommend this article to be published on Molecules at its current version.
Some of the main issues are as follows:
- Compared with the existing research, what are the main advantages of this synthesis method? There should be more specific explanation and introduction.
- This article mainly talks about the synthesis of some nanofibers, but it has only completed the characterization. In my opinion, there should be some application potentials described in the article to make it more specific. It should be published after supplementing the corresponding experiments and researches.
Some minor issues are as follows:
Line 66:How does the PVB benefit the enrichment of SeNPs?
Line136-137:How did this conclusion come about? Should there be some relevant references?
Author Response
The authors would like to thanks Reviewer 1 comments that improve our manuscript quality.
- We added more information with relevant references in the introduction.
- We added some sentences to the conclusions.
Point 1: The fiber-making process is traditional, but the polymer-enriching process is new. The novelty of the method that we developed is the way of producing nano selenium in a PVB solution and making fiber from a nanosuspension. The PVB is not the most common plastic that is used for electrospinning, and it has an important advantage: it is not water-soluble. The PVB is generally dissolved in 100 or 96% ethanol. We used 80% because the initial compounds (sodium selenite and ascorbic acid) do not dissolve well in pure alcohol. We found that after making the reaction we had to decrease the water content by adding water-free, alcoholic PVB solution for the proper electrospinning technique.
Point 2: We would like to share with other researchers the production methods for nano selenium-enriched nanofibers and their characterization. We are working on the testing of different applications, in this manuscript we wanted to mention only the possibility of filter and bandage application.
Point 3: We changed the sentence in the Results and Discussion Chapter. "Ten percent of PVB polymer was suitable for resuspension selenium nano powder"
Point 4: We added relevant references in sentences. "The applications of functional nanofibers are increasing in various technical fields including filtration [50,51], tissue engineering [3,7], drug delivery systems [12], wound dressings [21,23], and antibacterial [18,22]"
Reviewer 2 Report
The article entitles as, “Elemental Selenium Enriched Nanofiber Production” presents the preparation of red and grey selenium nanoparticles enriched nanofiber prepared via electrospinning technique form a polyvinyl butyral polymer solution. The authors have done a good job however they need to address following points before the consideration of publishing this article. My decision on the present form of this article is Major Revision.
- Include more details of electrospinning experimental parameters.
- The authors have reported the mean diameter of nanofibers (500 nm) while specific area and length are mentioned as their hypothesis. It is confusing and looks casual approach. It needs to be explained with proper characterization.
- Authors have fabricated the nanofibers on two substrates, it should be concluded that which substrate was better for fabrication.
- What are the advantages and disadvantages of bead formation?
- Red selenium nanogranules are amorphous while grey selenium nanogranules are crystalline according to XRD data. Does amorphous or crystalline nature of these nanogranules has any effect on morphology or any other property of nanofibers?
- It is suggested to include the SEM images of as synthesized red selenium and grey selenium nanogranules.
Author Response
The authors would like to thanks Reviewer 2 for comments that improve our manuscript quality.
Point 1: We included process and system parameters of electrospinning in the relevant sections. "Flow rate - 10 mL/h; Distance between stain-less-steel collector and needle tip - 25 cm; Voltage - 40 kV; concentration of polymer - 10%; solvent - ethanol; Voscosity - 474 mm2/s; Density - 0.81 g/mL; Humidity - 34%; Temperature - 24 0C"
Point 2: The surface area was calculated for a fiber, which has a smooth surface. According to the SEM pictures, the nanofiber which contained nano selenium only inside was exactly like this. We consider our simple calculation gives a good estimation not only for the length but for the surface area as well. When the selenium is outside, then the estimation is good for the length, but not good for the surface area. In this case, our calculation gives an estimation of only the minimum surface area.
Point 3: Our aim was to produce nanofiber that contains the nanoparticle only inside. We consider it has some advantages, but it requires further study, what we would like to publish later.
Point 4: The advantages of bead formation are to be load more selenium in sheet and increase its releasing and positive effects on the surface. Due to these advantages, red and grey selenium nanopowder embedded nanofibers are more suitable for filtration design in water and air and wound healing processes. But the chain-like structure of beads may affect fiber's strength.
Point 5: There was no difference in the diameter size or morphology of the nanofibers, and electrospinning process between amorphous and crystalline forms of selenium.
Point 6: Red selenium nano granules as amorphous and grey selenium nano granules as crystalline in shape were confirmed by X-ray diffraction analyses. The SEM pictures did not show a difference.

Round 2
Reviewer 1 Report
This article introduced a method of using Electrospinning to enrich SeNPs and synthesis relevant nanofiber. Generally speaking, the article has some innovation and value, but there are some issues to be improved in the article. So I recommend that this article can be published on Molecules after the corrections.
Some issues are as follows:
- Line 75: How does it came to the expression “As a result”? Is the conditions of the electrospinning process came from the Reference 46? There should be explanations.
- Line 78:“The study reported…”is not the common expression, I think it should be “It is reported that…”.
- Line 130:How does it come to the conclusion “80 % of ethanol was not good for electrospinning”? Should there be some relevant references?
- Line 161-162: “For instance, the addition of selenium nanoparticles in nanofibers were inhibited bacteria by reducing bacterial cell activity [3,],” the reference should be marked clearly.
Author Response
Response to Reviewer 1
The authors would like to thank Reviewer 1 for comments that improve our manuscript quality.
- Line 75: How does it came to the expression “As a result”? Is the conditions of the electrospinning process came from the Reference 46? There should be explanations.
- That meant our results were starting. So, we changed "As a result" to "In our result".
- Line 78: “The study reported…”is not the common expression, I think it should be “It is reported that…”.
- We changed this sentence by “It is reported that all solutions based on isopropanol, butanol and ethanol were good solvents for PVB in the production of nanofiber [39].”
- Line 130: How does it come to the conclusion “80 % of ethanol was not good for electrospinning”? Should there be some relevant references?
- There is no relevant reference. It is our conclusion. We observed it while electrospinning.
- Line 161-162: “For instance, the addition of selenium nanoparticles in nanofibers were inhibited bacteria by reducing bacterial cell activity [3,],” the reference should be marked clearly.
- We explained the sentence. “For instance, the addition of selenium nanoparticles in nanomaterials improved fibroblast’s metabolic activity while reducing the ATP content of S. aureus [3] and completed re-epithelization [23] for skin application”.

Reviewer 2 Report
Revised version is enough for publication.
Author Response
The authors would like to thank Reviewer 2 for considering our manuscript. Your comments were very helpful to improve our manuscript quality.